# How do people feel about sleep diaries? Factors influencing sleep diaries completion with and without daily exposure to light

Boris Gass[1,2]*, Grégory Pierné[1], Elisabeth Ruppert[1], Ulker Kilic-Huck[1], Laurence Hugueny[1], Patrice Bourgin[1,3], Juliette Chambe[1,2,4]

1 Sleep Disorders Center & CIRCSom (International Research Center for ChronoSomnology), Strasbourg University Hospital, 1 place de l'hôpital, Strasbourg, France, 2 General Medicine Department, Faculty of Medicine, University of Strasbourg, Strasbourg, France, 3 C7NRS UPR, Institute for Cellular and Integrative Neurosciences, 5 rue Blaise Pascal, Strasbourg, France, 4 Neuhof Multi-professional University Health Center (MSPU du Neuhof), Strasbourg, France

* juliette.chambe@unistra.fr

**Data availability statement:** The data underlying the results presented in the study are available online : https://osf.io/9vqms/?view_only=165f81f6ca06434da04870d483909fd6

**Funding:** The author(s) received no specific funding for this work.

## Abstract

### Study Objectives

Sleep diary is a common tool in sleep medicine, but was barely validated in practice so far. Lack of light exposure is associated with sleep disorders, especially insomnia. For that purpose, our team developed a sleep diary with information about light exposure. Our objectives were to compare the proportion of informative usual sleep diaries (USD) and light/sleep diaries (LSD) and to evaluate factors influencing the quality of filling them.

### Methods

A monocentric, prospective, controlled interventional study was conducted. Patients included were randomized into two parallel groups: distribution of an USD (used in common practice), versus distribution of a LSD (similar to the USD but with light exposure data). The main outcome was the proportion of diaries returned and correctly filled out. A multivariate logistic regression model was then used to identify factors associated with the correct filling.

### Results

A total of 325 USD and 324 LSD were distributed. 295 (45.5%) diaries were returned by patients: 158 (48.6%) USD, 137 (42.3%) LSD. The proportion of correctly completed diaries was 25.2% for USD versus 20.4% for LSD, which corresponds to a difference in the proportion of −4.86% [−10.25%; +∞[. The hypothesis of non-inferiority of LSD compared to USD cannot therefore be retained for a non-inferiority threshold of −10%. Multivariate analysis identified the level of difficulty experienced by the patient as an important factor influencing the quality of completion (OR = 0.99 [0.979–0.998]). The proportion of returned sleep diaries was greater in the USD and LSD insomniac sub-group (p < 0.0001).

**Competing interests:** The authors have declared that no competing interests exist.

**Abbreviations:** USDusual sleep diaryLSDlight/sleep diaryVASvisual analog scaleINSEEthe French National Institute for Statistics and Economic StudiesCNILthe French National Commission for Information and LibertiesORodds ratio95% CI95% confidence intervals.

## Conclusions

The non-inferiority of this new diary was not reached but this study highlighted the importance of clear explanations, patient motivation and restricted amount of collected data. Interestingly, insomniac patients represent a major target population for this tool.

## Introduction

Light plays an essential role in the regulation of sleep [1]. Field studies of populations subjected to low levels of lighting demonstrated links between low light exposure (artificial lighting in domestic, commercial or industrial environments, indoor work) and various sleep disorders (circadian rhythm disorders with advanced or delayed sleep-wake phase, irregular sleep-wake rhythm) [2]. Low daytime light exposure could lead to sleep disturbances in humans, particularly insomnia [3]. The hypothesis that a lack of lighting or inadequate lightning exposure may contribute to sleep disturbances in older patients is supported by the effectiveness of light therapy [4]. Identifying populations of insomniacs with the lowest exposure to daylight could therefore be of interest in current therapy.

The luxmeter, often coupled with the actimeter, has been used for this purpose [5]. This device commonly takes the form of a wristwatch and is discreet for the patient. Still, it does not enable illuminance to be measured at eye level, resulting in measurement error. Even though actimeters provide more objective and reliable data than sleep diaries, these devices are expensive and not widely accessible in routine clinical practice. An alternative approach would be to record this information in a sleep diary [6]. The sleep diary is one of the most widely used tool in sleep medicine. [7–10]. The patients have usually to complete a paper sleep diary for a defined period of time [10]. They record in the sleep diary the timing of their sleep and bedtime periods, based on their subjective recollection of the previous night. Sleep diaries allow to collect information on real-life sleep patterns and provide a simple measure of the circadian sleep-wake rhythm. These diaries are commonly used for diagnostic purposes (insomnia, circadian rhythm sleep-wake disorders), but also for therapeutic follow-up [8]. Usual data collected are bedtime/lights-out time, time falling asleep, wake-up and getting up times, duration and number of awakenings during sleep, duration and number of naps. Analyzing the diary enables to estimate the latency to sleep onset, total sleep time, sleep efficiency, time spent awake after sleep onset [11]. Moreover, the sleep log provides great information on the circadian phase of the sleep-wake cycle and chronotype of the subject by estimating the mid-sleep time from sleep onset and sleep offset time. Therefore, the diary is useful to target the symptoms reported by the patient and to refine the diagnosis and severity of the sleep disorder. Data of sleep diaries are complementary to the data collected by actimetry or polysomnography [9,12]: it allows for a subjective measurement of sleep, which can be compared to the objective data provided by an actimeter or polysomnography. There is a discrepancy between objective measures of sleep and data collected by sleep diary; those discrepancies also vary from healthy people to patients with sleep disturbances, such as insomnia. Therefore, the diary plays a crucial role in patient care because these subjective data reflect the patient's experience, especially for insomnia[13–15].

Additional data can be collected, targeted on the specificity of each patient (e.g., perceived sleep quality, alcohol, caffeine, tabacco consumption, medication, exercise)(16). However, there is currently no validated diary that facilitates the collection of light exposure data. In 2013, the Sleep Disorders Department developed a modified sleep diary to collect data related to light exposure in addition to the usual information on waking-sleeping patterns. Both types of diaries have been commonly used in the department ever since. Due to the additional

amount of information required by patients to collect, this new tool is more complex compared to the diary currently used. As a result, there is doubt as to its ability for reporting sleep habits with metrological characteristics compared to those of the usual tool.

Our objective was to estimate the proportion of returns from the two diaries and to see if the proportion of correct filling of the usual sleep diary (USD) could be no less than that of the light/sleep diary (LSD). The secondary objective was to highlight the factors associated with the correct filling of the two diaries.

## Methods

A monocentric, prospective, controlled interventional study was conducted on 649 patients between September 27, 2016 and June 30, 2017.

### Tools used

Two types of sleep diaries were used in this study. The USD (Fig 1A), used in common practice in a sleep department, was in the form of a grid. The hours of the day were in columns and the days of the month were in rows. The patient reported the time of day (with vertical arrows pointing up and down), the time of night (hatched rectangle), and the time of day (vertical arrows pointing down and up), respectively.

The light/sleep diary (LSD) (Fig 1B), an innovative tool, is the result of a test phase of seven successive versions of the diary, which initially included 5 light levels. Feedback from patients and medical teams has led to a simplified consensual version: it distinguishes between exposure to natural light outdoors, exposure to artificial light indoors, and exposure to darkness. The LSD was similar to the previous one, but additionally included periods of outdoor daylight exposure (unshaded rectangles) and hours spent in the dark (solid horizontal lines delimited by thick dots).

These two diaries have a precision of a quarter of an hour and are accompanied by an explanatory leaflet and a questionnaire to record certain variables of interest. Both diaries incorporate the elements from the consensus sleep diary[7], while maintaining the more visual graphic aspect for clinical practice.

Each diary was randomly assigned to either USD or LSD, with its own identification number. It was distributed either by mail by the department secretary or given by the nurse in the ambulatory sector along with explanations on how to complete it. A register of distributed diaries was established. They were asked to fill it out daily until their next consultation.

### Data collection

The patients included were randomized in two parallel groups that were considered independent: distribution of an USD versus distribution of a LSD.

To be included, patients had to be of legal age, had a consultation or hospitalisation in the sleep pathology department of Strasbourg University Hospital, and provided written consent. Patients with physical or cognitive disabilities resulting in an inability to complete the diary were not included.

For each type of diary, the main outcome was the proportion of diaries that were returned and correctly completed relative to the total number distributed.

In the absence of a reference to consider whether an agenda was correctly completed, the research team established a consensus according to the following criteria: at least one line had to be dated and at least 7 consecutive days were completed. Each line corresponding to one day was to have a pair of arrows pointing in opposite directions. A hatched section was to have clear boundaries that would allow accuracy to within 15 minutes. Any other symbol or

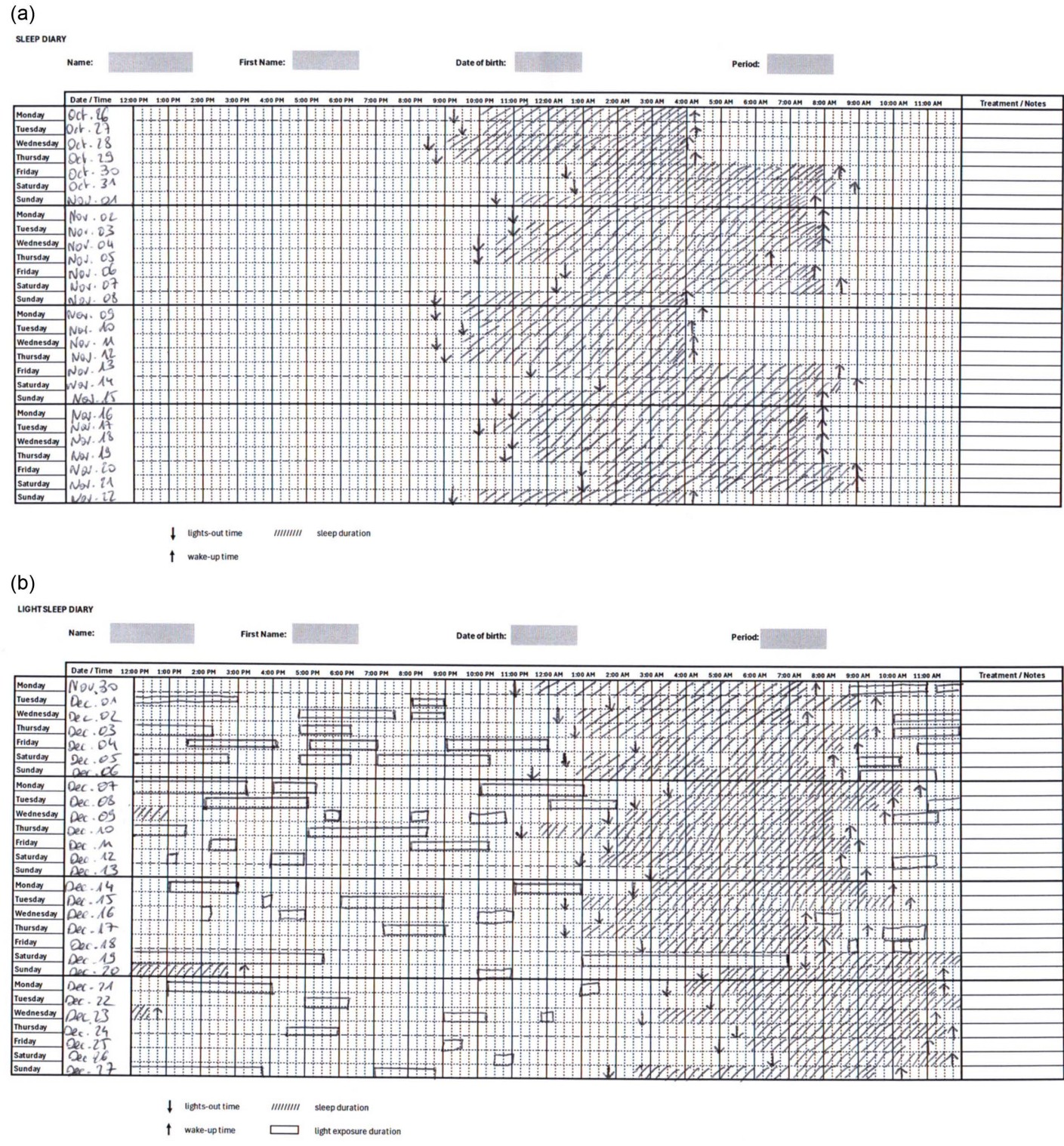

**Fig 1. Presentation of the sleep diaries.** (A) The usual sleep diary (USD) – The downward arrow marks the lights-out time, the upward arrow indicates the wake-up time, and the hatch marks represent the sleep duration. (B) The light/sleep diary (LSD) – The downward arrow marks the lights-out time, the upward arrow indicates the wake-up time, and the hatch marks represent sleep duration. Unshaded rectangles indicate periods of outdoor daylight exposure.

text not provided for in the instructions and which could interfere with interpretation should not be present on the grid. At the same time, the questionnaire attached to the diary was used to record the patient's age, sex, socio-professional group (INSEE categories), level of diploma (INSEE categories), subjective difficulty in completing the diary using a 100 mm visual analogue scale (VAS), and history of sleep pathologies grouped according to the major categories of the ICSD-3.

## Data analysis

For the analysis, patients who returned their diary were divided into two groups according to the distributed diary, namely USD or LSD.

For a first species alpha risk at 5%, a study power at 80%, a predicted correctly completed diary rate at 50% and a chosen non-inferiority threshold at 10%, the number of diaries to be distributed was estimated at 350 distributed diaries per group.

The characteristics of the two patient populations (USD versus LSD group) were first described. Quantitative variables were presented according to their means and standard deviations (if normally distributed), if not according to their median and interquartile range. Qualitative variables were presented by numbers (percentages) for each modality.

In order to verify the non-inferiority conditions, we presented the analysis as per protocol and then as intention to treat.

To meet the secondary objective, we first sought the links, through univariate analysis, between the proportion of correctly completed agendas (variable to be explained) and the risk factors studied (explanatory variables) using Mann–Whitney Wilcoxon tests, Chi2 tests or exact Fisher tests depending on the conditions of application of the present tests. A multivariate logistic regression model was then used, incorporating the risk factors from the univariate analysis for which the p-value was less than 0.15 according to a descending stepwise selection of variables. Finally, we looked for the distribution by pathology of patients consulting or hospitalized in the department in order to compare it with that of patients who returned the diary.

## All the analyses were carried out using R software version 3.1

**Regulatory and ethical aspects.** This project was the subject of a declaration to the French National Commission for Information and Liberties (CNIL) and an entry in the register of computer processing and liberties of the University of Strasbourg. We collected written informed consent of patients participating in the study. It received a favourable opinion from the Ethics Committee of the University Hospitals of Strasbourg (No. FC/2015-74).

## Results

A total of 649 sleep diaries were distributed between September 27, 2016 and June 30, 2017: 325 USD, 324 LSD. As of June 30, 2017, 295 diaries were returned by patients: 158 USD, 137 LSD. Table 1 shows the patient characteristics according to the returned diary.

They are mostly women, with an average age of 52. Patients are predominantly employed or retired. Nearly half of the responding patients have a bachelor's degree. Insomniac patients represent the second most frequent pathology in the sample in terms of frequency (19.9% and 22.6% respectively) behind respiratory pathologies (45.2 and 49.4% respectively), whereas there is no difference in the distribution of patients coming for respiratory disorders, circadian rhythm disorders, hypersomnia and parasomnia. Some patients had more than one diagnosis of sleep pathology. We compared this distribution to that of patients seen in consultation and hospitalization in the sleep department of Strasbourg

**Table 1. Baseline characteristics of patients who have returned the usual sleep diary (USD) or the light/sleep diary (LSD).**

| Characteristic | USD (n = 158) | LSD (n = 137) |
|---|---|---|
| **Mean age (SD)** | 52.5 (16.6) | 52.2 (15.9) |
| **Female, n (%)** | 82 (51.9) | 74 (54.0) |
| **Socio-professional category, n (%)** | | |
| Craftsman, merchant or company manager | 2 (1.3) | 2 (1.5) |
| Framework | 19 (12) | 14 (10.2) |
| Employee | 35 (22.2) | 25 (18.2) |
| Intermediate profession | 6 (3.8) | 7 (5.1) |
| Worker | 14 (8.9) | 7 (5.1) |
| Retired | 38 (24.1) | 38 (27.7) |
| No professional activity | 21 (13.3) | 19 (13.9) |
| No answer | 23 (14.6) | 25 (18.2) |
| **degree, n (%)** | | |
| Baccalaureate | 17 (10.8) | 19 (13.9) |
| Baccalaureate degree + 2 | 20 (12.7) | 23 (16.8) |
| 2nd or 3rd university cycle, grande école | 28 (17.7) | 23 (16.8) |
| vocational training qualification | 49 (31.0) | 31 (22.6) |
| Without a college diploma or certificate | 19 (12.0) | 18 (13.1) |
| No answer | 25 (15.8) | 23 (16.8) |
| **Sleep pathologies, n (%)** | | |
| Respiratory disorders | 84 (45.2) | 83 (49.4) |
| Central hypersomnia | 5 (2.7) | 6 (3.6) |
| Insomnia | 37 (19.9) | 38 (22.6) |
| Circadian rhythm disorders | 13 (7.0) | 9 (5.4) |
| Parasomnia | 3 (1.6) | 6 (3.6) |
| Abnormal movements | 31 (16.7) | 19 (11.3) |
| Under exploration | 13 (7.0) | 7 (4.2) |

SD, standard deviation; n, number; %, percentage of total population.

University Hospital between 1 and 31 October 2016: out of 346 patients studied, it was found that the distribution by pathology was globally very close to that of patients having returned their diary, except for insomniac patients who were twice as represented among patients having returned their diary than among the usual patients of the department (21.2% vs 9.2%, p < 0.0001) (Table 2).

The proportion of diaries returned (correctly and not correctly filled out) was 45.5% of the diaries distributed on average and slightly higher for the usual diaries (48.6% and 42.3% respectively for USD and LSD) (Table 3).

They were completed over 20 days in 77.7% of cases for USD and 63.0% of cases for LSD. The proportion of diaries returned correctly completed was 25.2% for USD versus 20.4% for LSD, which corresponds to a difference in the proportion of correctly completed diaries of −3.72 [−13.2; +∞] in per protocol analysis and −4.86% [−10.25%; +∞[ in intention-to-treat analysis. The hypothesis of non-inferiority of LSD compared to USD cannot therefore be retained for a non-inferiority threshold of −10%.

In order to better understand the observed proportions, we then studied the factors associated with the correct filling of the diary (Table 4).

**Table 2. Distribution of the patients by disease, Department of Sleep Diseases, Strasbourg, october 2016.**

| Disease | Study (%) | Department (%) | p[1] | Post-hoc [1] |
|---|---|---|---|---|
| Respiratory disorders | 167 (47.2%) | 173 (50.0%) | <0.0001 | 0.65 |
| Central hypersomnia | 11 (3.1%) | 24 (6.9%) | | 0.041 |
| Insomnia | 75 (21.2%) | 32 (9.2%) | | <0.0001 |
| Circadian rhythm disorders | 22 (6.2%) | 18 (5.2%) | | 0.38 |
| Parasomnia | 9 (2.5%) | 5 (1.4%) | | 0.22 |
| Abnormal movements | 50 (14.1%) | 37 (10.7%) | | 0.066 |
| Under exploration | 20 (5.6%) | 91 (26.3%) | | <0.0001 |

%, percentage of total population;

[1]global Chi-square test; p, p-value.

**Table 3. Numbers of patients according to the agenda and whether or not they are returning, n (%).**

| | USD | | | LSD | | | All diaries | | |
|---|---|---|---|---|---|---|---|---|---|
| Correct | 82 (25.2) | 158 (48.6) | 82 (25.2) | 66 (20.4) | 137 (42.3) | 66 (20.4) | 148 (22.8) | 295 (45.5) | 148 (22.8) |
| Not correct | 76 (23.4) | | 243 (74.8) | 71 (21.9) | | 258 (79.6) | 147 (22.6) | | 501 (77.2) |
| Not returned | 167 (51.4) | 167 (51.4) | | 187 (57.7) | 187 (57.7) | | 354 (54.6) | 354 (54.5) | |
| Total | 325 (100) | | | 324 (100) | | | 649 (100) | | |

Correct, number of correctly filled diaries; Not correct, number of diaries not correctly filled out; Not returned, number of unreturned agendas; USD, usual sleep diary; LSD, Light/Sleep Diary; %, percentage of total population.

In univariate analysis, the distribution of the diary by a secretary was significantly related to poorer diary completion (OR = 0.47 [0.29–0.76]). The high level of subjective difficulty in completing the agenda evaluated by the VAS also increased the return of not correctly completed agendas (OR = 0.99 [0.98–0.99]). Conversely, giving explanations when the agenda was distributed would double the chances of obtaining a better completed agenda (OR = 1.96 [1.17–3.29]). Parasomnia was the sleep disorder most strongly associated with the return of an improperly completed diary, although this association was not significant (OR = 0.27 [0.04–1.16]). On the other hand, age, the type of diary distributed (USD or LSD), the socio-professional category to which the patient belongs, the patient's level of education, or the other sleep disorders studied were not associated with a difference in the quality of diary completion.

Finally, in multivariate analysis, only the increase in the level of difficulty experienced by the patient when completing the diary was significantly associated with a decrease in the proportion of correctly completed diaries (OR = 0.99 [0.979–0.998]) (Table 5).

## Discussion

We wanted to compare two sleep diaries used in current practice in a sleep pathology department of a university hospital centre. The idea was to investigate a simple tool for estimating light exposure under real-life conditions by modifying the usual sleep diary, familiar to caregivers and patients with sleep disorders, by adding light exposure data. The results of the study did not show the non-inferiority of this new LSD compared to USD. The study of the factors associated with the correct filling of the two diaries identified as the only significant factor the level of difficulty felt in filling it.

The aim of this study was not to validate the use and the reliability of the LSD, already a daily practice for more than four years in the Sleep Disorders Department, with a clinical

**Table 4. Univariate analysis of the factors associated with the correct filling of all agendas (n = 295).**

| Factors under consideration | Correctly completed | | OR | 95% CI | p |
|---|---|---|---|---|---|
| | yes | no | | | |
| **Age (mean, SD)** | 52.18 (16.2) | 52.6 (16.3) | 0.99 | [0.98–1.01] | 0.77[1] |
| **Sex (ref = female)** | 148 | 142 | 0.7 | [0.44–1.11] | 0.13[2] |
| **Agenda (ref = usual agenda)** | 148 | 147 | 0.86 | [0.54–1.36] | 0.52[2] |
| **Distribution of the agenda (ref = secretary)** | 148 | 147 | **0.47** | **[0.29–0.76]** | **0.01[2]** |
| **Explanations** | 126 | 114 | **1.96** | **[1.17–3.29]** | **0.01[2]** |
| **Difficulty felt (VAS) (median, IQR)** | 20 (4–50) | 38(10–64.5) | **0.99** | **[0.98–0.99]** | **0.01[1]** |
| **Socio-professional category, n (%)** | | | | | |
| Craftsman, merchant or company manager | 1 | 3 | 0.27 | [0.01–2.21] | |
| Framework | 14 | 19 | 0.6 | [0.26–1.35] | |
| Employee | 35 | 25 | 1.13 | [0.57–2.25] | |
| Intermediate profession | 9 | 4 | 1.82 | [0.54–7.19] | |
| Worker | 10 | 11 | 0.73 | [0.28–1.95] | |
| Retired | 42 | 34 | 1.00 | Ref | |
| No professional activity | 18 | 22 | 0.66 | [0.30–1.42] | 0.42[3] |
| **degree, n (%)** | | | | | |
| Baccalaureate | 20 | 16 | 1.00 | Ref | |
| Baccalaureate degree + 2 | 26 | 17 | 1.22 | [0.50–3.02] | |
| 2nd or 3rd university cycle, high school | 24 | 27 | 0.71 | [0.29–1.67] | |
| vocational training qualification | 46 | 34 | 1.08 | [0.49–2.39] | |
| Without a college diploma or certificate | 14 | 23 | 0.49 | [0.19–1.23] | 0.21[2] |
| **Sleep pathologies, n (%)** | | | | | |
| Respiratory disorders | 89 | 78 | 1.33 | [0.84–2.12] | 0.22[2] |
| Central hypersomnia | 6 | 5 | 1.2 | [0.35–4.25] | 0.77[2] |
| Insomnia | 38 | 37 | 1.03 | [0.61–1.74] | 0.92[2] |
| Circadian rhythm disorders | 9 | 13 | 0.67 | [0.27–1.60] | 0.37[2] |
| Parasomnia | 2 | 7 | **0.27** | **[0.04–1.16]** | **0.10[3]** |
| Abnormal movements | 27 | 23 | 1.2 | [0.65–2.23] | 0.55 [2] |
| Under exploration | 12 | 8 | 1.53 | [0.61–4.02] | 0.36 [2] |

OR, odds-ratio; 95% CI, 95% confidence interval; p, p-value; VAS, visual analog scale IQR, interquartile rank;

[1]Mann Whitney Wilcoxon test;

[2]Chi-square test;

[3]Fisher test.

**Table 5. Multivariate analysis of factors associated with the correct filling of all agendas, logistic regression.**

| Factors | Adjusted OR | 95% CI | p |
|---|---|---|---|
| **Sex** | 0.91 | [0.495–1.663] | 0.76 |
| **Distribution** | 0.82 | [0.355–1.924] | 0.65 |
| **Explanations** | 1.26 | [0.551–2.913] | 0.59 |
| **VAS** | 0.99 | [0.979–0.998] | 0.02[*] |
| **Parasomnia** | 0.20 | [0.010–1.463] | 0.16 |

OR, odds-ratio; 95% CI, 95% confidence interval; p p-value;VAS, visual analog scale.

reliability estimated by the practitioners. Instead, the focus was on determining whether introducing light exposure information affected the return rate of the diaries.

The main outcome was the proportion of agendas completed "satisfactorily". It was necessary to establish a set of rules to discriminate agendas judged to be of acceptable quality for clinical interpretation, as these criteria had never been described until now. These rules were the subject of a consensus within the service. A minimum duration of 7 days was chosen to consider the agendas as valid. This duration corresponds to the minimum duration proposed by ICSD-3 each time it invites the use of the diary to establish a diagnosis [16]. However, the recommended duration is 14 days, as the 7-day duration is too short to reveal a number of variations in patients' sleep patterns. A data quality control was performed by checking the answers of the questionnaire attached to the diary with the patient label on 4 items (surname, first name, date of birth, gender). To limit the risk of incomplete returns, the questionnaires and diaries were stapled together, marked with a unique number, and patient identified on both documents. The variables with the highest risk of error were: the socio-professional group, when belonging to a given category was not obvious, and the visual analogue scale. The latter was the variable for which the greatest number of erroneous and missing data were found (31.6% were missing). This may have been related to a difficulty in understanding the instruction due to the unusual nature of the check boxes. A Likert scale might have been able to limit this effect.

## Limitations of the study

The a priori calculation of the number of patients required involved 350 diaries per group, assuming that 50% of the diaries were returned and filled satisfactorily. In the absence of data on the subject in the literature, this value was proposed on the basis of our clinical judgement. This figure seems to have been accurately estimated since out of 295 diaries returned, 148 (50.2%) were completed according to the criteria. We considered a 9-month follow-up period adequate given the high activity at the consultation center. While this did not achieve the expected number of subjects, it came close (649 diaries were actually distributed versus the 700 expected). The return rate of 45.5% had not been anticipated and is responsible for the unmet requirement. The non-inferiority threshold for the main objective was –10%. This value was chosen arbitrarily in the absence of literature to guide the decision. It is a good compromise between a sufficiently small difference so that clinical comparability can be maintained between the two agendas and an acceptable number of patients to be included in order to conclude. A non-inferiority threshold of 5% would have required 350 to 1,000 patients per group, an even more difficult goal to achieve, given that it took more than six months to distribute these 649 diaries. In the case of a non-inferiority trial, the per protocol analysis (with exclusion of unreturned diaries) is theoretically to be preferred, because it allows us to place ourselves in the maximum hypothesis of showing a difference [17]. The intention-to-treat analysis (with inclusion of diaries not returned by patients as well as diaries not correctly filled in) is biased because it reduces the risk of showing a difference, and is therefore better suited to a superiority trial [18]. We chose to perform both analyses because the inclusion of unreturned diaries had an important practical significance: the diary being the basic tool for most sleep pathologies, the inclusion of patients who do not complete the diary is critical. It was not possible to conclude that the new diary was not inferior to the previous one, either in the intention-to-treat analysis or in the per-protocol analysis. This can be explained by the low proportion of returned agendas, which we had not anticipated and leads to a lack of power of our study. In spite of this, this result is not surprising. Indeed, the addition of light exposure data adds complexity to the diary, which can be a brake on its filling. This confirms the clinical impression of keeping a tool as simple as possible, and

of targeting the use of a more complex diary in specific situations [19]. Although there is probably little difference between the two diaries in the proportion of correct filling, the burden of filling the diary should be reduced in order to achieve better compliance [20]. This could be achieved by limiting the time for filling the diaries, but it would also be possible to better target patients to whom the diaries should be offered by offering them to those who feel the need for them and do not perceive them as redundant. All this also requires oral explanations when the diary is handed over. That less than half of the agendas were returned was a real surprise and led us to question and hypothesize the causes of this result. Although a high rate of return of these diaries is ideal, in reality, adherence to medical instructions by patients is generally estimated to be around 50%, which significantly varies due to chronic illnesses. This study shows that the return rate of diaries is close to this average, with no significant difference between USD and LSD. Multiple factors contribute to this shortfall: the complexity of some diaries can make daily completion burdensome, and the requirement for regular updates might deter consistent use over several weeks. Certain requirements could also pose data privacy concerns. Instructions are also sometimes lacking, which could further hinder diary compliance and completeness after distribution. Clearer explanations or periodic reminders could motivate patients who might not fully understand the importance of diary compliance. Regular follow-up might encourage higher diary return rates and provide valuable data for further investigation. All these elements are important levers for improving diary return rates in future studies. The identification of the specific characteristics of those lost to sleep was not possible due to the data collection that took place when the diaries were returned. These data are not known for patients who did not return the distributed diary. Another hypothesis is that of targeting patients to whom it should be distributed: Patients who were more engaged with their treatment might have been more likely to return their diaries. Motivation appears to be a key factor in completing the sleep diary. Insomniac patients were proportionally twice as represented in the population, having returned the diaries compared to the population seen in the sleep department during this period. This reinforces our impression that this population, which is one of the main targets of our agenda, found the filling of this agenda useful[15,21]. We can also hypothesize that certain factors evaluated for the quality of agenda completion influenced this return rate.

We thus made the hypothesis that the factors associated with correct filling were not different from one schedule to another. The analysis of these factors was done by considering the overall number of diaries returned, regardless of the type of diary, in order to increase the power of the results. The univariate analysis showed that the quality of filling was better when the diary was given in person and with explanations, and when it was deemed easier to fill. In multivariate analysis, we suspected an interaction between the variables: distribution methods, explanations, subjective difficulty felt. Indeed, the nurses' interventions accompanied by explanations with patients could have had an effect both on the quality of the information collected and on the impression of difficulty felt when filling in the diary. Effect not found when the diaries were distributed by the secretaries by post. During the construction of the model, no significant interaction model tested was found. This nevertheless led to a questioning of the way the service operates for the distribution of this tool: patients' adherence to an assessment and support tool, whatever the pathology, depends on their understanding of it and the way they take it on board [22]. In the evaluation of the factors associated with a correct filling of the diary, we had hypothesized a better filling for patients with a higher level of education and related socio-professional categories. This was not the case. This result is interesting because it suggests that it is a tool that would not deepen social inequalities in health. It also raises the hypothesis that its completion is more related to patient motivation than to the understanding of the tool. In fact, when the diaries were correctly filled out, we found that they were often

filled out over a longer period than recommended, several weeks. This element confirms the good appropriation of the tool by these patients, and shows the interest they can find in it.

In current practice, diaries are used to try to limit recall bias phenomena encountered when information on past events needs to be gathered retrospectively from simple patient-clinician interviews by recording the event at the time of its occurrence [23]. However, these agendas require repeated recording, which is the reason for the resulting compliance problem [20]. The problem of concealment of data must also be addressed. A study by Mazze et al. found that 2/3 of the patients had recorded false values to mask hypo- or hyperglycaemia in their blood glucose diary [24].

At present, electronic diaries are available for use with smartphones or tablets, and a sleep diary for patients, doctors and researchers is in the development phase [25]. They would have the advantage of time-stamping the data and thus limit the memory bias to improve compliance [26]. For the sleep diary, it would reduce reading ambiguities related to handwritten entries, limit interpretation time during consultations, and provide easier access to the diary [22]. Tonetti et al. compared an electronic tablet diary with a paper diary and actimetry in 15 healthy volunteers. The authors concluded that the two forms of the diary were equivalent and that there was a practical advantage for the electronic diary. Some studies also support the feasibility of estimating light exposure with a diary tool [27].

The use of the diary could lead to better compliance and greater ease of use and availability of the diary.

In conclusion, this study introduced a modified sleep diary to report usual lighting conditions in order to identify situations at risk of light exposure deficiency. The non-inferiority of this new diary compared to the traditional diary concerning the quality of filling could not be shown by a lack of power resulting from a large number of lost sight. Nevertheless, the targeting of patients, the provision of explanations, the limitation of information and the switch to electronic support offer interesting prospects for improving the return and correct filling rates of the distributed diaries. Future studies could be conducted on a larger scale, involving additional centers over a longer study period. Focusing on specific sleep troubles, especially chronic insomnia and circadian rhythm troubles should improve the patients' motivation. Furthermore, follow-up campaigns, a hybrid format (electronic or paper), clear instructions at the time of distribution, and verification of understanding could help achieve a better return rate. Several additional studies could gradually validate the new LSD diary.

## Acknowledgments

The authors are grateful to the staff of the Sleep Disorders Center of the Strasbourg University Hospital for their contribution.

## Author contributions

**Conceptualization:** Grégory Pierné.

**Data curation:** Grégory Pierné, Elisabeth Ruppert.

**Formal analysis:** Boris Gass, Grégory Pierné.

**Investigation:** Grégory Pierné, Ulker Kilic-Huck.

**Methodology:** Elisabeth Ruppert, Ulker Kilic-Huck, Patrice Bourgin, Juliette Chambe.

**Project administration:** Grégory Pierné, Laurence Hugueny.

**Resources:** Laurence Hugueny, Patrice Bourgin.

**Supervision:** Elisabeth Ruppert, Patrice Bourgin, Juliette Chambe.

**Validation:** Juliette Chambe.

**Writing – original draft:** Boris Gass.

**Writing – review & editing:** Juliette Chambe.

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
