## [Decision Letter · Decision Letter 0]

25 Jun 2024

PONE-D-23-38179How do people feel about sleep diaries? 

Factors influencing sleep diaries completion with and without daily exposure to lightPLOS ONE

Dear Dr. Chambe,

Thank you for submitting your manuscript to PLOS ONE. After careful consideration, we feel that it has merit but does not fully meet PLOS ONE’s publication criteria as it currently stands. Therefore, we invite you to submit a revised version of the manuscript that addresses the points raised during the review process.

We look forward to receiving your revised manuscript.

Kind regards,

Julio Alejandro Henriques Castro da Costa

Academic Editor

PLOS ONE

Journal Requirements:

2. Please include a separate caption for each figure in your manuscript.

Reviewers' comments:

Reviewer's Responses to Questions

**Comments to the Author**

1. Is the manuscript technically sound, and do the data support the conclusions?

Reviewer #1: Yes

2. Has the statistical analysis been performed appropriately and rigorously? 

Reviewer #1: Yes

3. Have the authors made all data underlying the findings in their manuscript fully available?

Reviewer #1: Yes

4. Is the manuscript presented in an intelligible fashion and written in standard English?

Reviewer #1: Yes

5. Review Comments to the Author

Reviewer #1: The study provides valuable insights into the use and improvement of sleep diaries, especially regarding the practical application of incorporating light exposure information. However, there is room for improvement in the background introduction, methodological details, and discussion of results. I suggest the authors consider the following points to enhance the quality and impact of the article:

The study does not clearly explain how the reliability of the sleep diaries was validated.

The content and results of the study are somewhat unconvincing, especially given the low return rate of the diaries.

It is recommended to increase the discussion on the limitations of the study, such as insufficient sample size and short follow-up duration, and propose directions for future research improvements.

Ensure the uniformity and accuracy of citation formatting

6. PLOS authors have the option to publish the peer review history of their article (what does this mean?). If published, this will include your full peer review and any attached files.

Reviewer #1: No

---

## [Author Response · Author response to Decision Letter 1]

20 Aug 2024

Dear editor,

We would like to thank you for your insightful feedback on our study. We are pleased to hear that you find our work on the use and improvement of sleep diaries valuable, particularly regarding the practical application of incorporating light exposure information. We appreciate your suggestions for enhancing our manuscript and have addressed your comments.

Please find enclosed our revised manuscript with developed introduction, methods and discussion sections.

We believe these revisions have strengthened our manuscript and addressed the concerns raised. Thank you again for your feedback and for the opportunity to improve our work. We look forward to your further comments and hope that our revised manuscript meets your expectations.

---

## [Decision Letter · Decision Letter 1]

14 Oct 2024

PONE-D-23-38179R1How do people feel about sleep diaries? Factors influencing sleep diaries completion with and without daily exposure to lightPLOS ONE

Dear Dr. Chambe,

Thank you for submitting your manuscript to PLOS ONE. After careful consideration, we feel that it has merit but does not fully meet PLOS ONE’s publication criteria as it currently stands. Therefore, we invite you to submit a revised version of the manuscript that addresses the points raised during the review process.

We look forward to receiving your revised manuscript.

Kind regards,

Julio Alejandro Henriques Castro da Costa

Academic Editor

PLOS ONE

Journal Requirements:

Reviewers' comments:

Reviewer's Responses to Questions

**Comments to the Author**

1. If the authors have adequately addressed your comments raised in a previous round of review and you feel that this manuscript is now acceptable for publication, you may indicate that here to bypass the “Comments to the Author” section, enter your conflict of interest statement in the “Confidential to Editor” section, and submit your "Accept" recommendation.

Reviewer #1: All comments have been addressed

Reviewer #2: (No Response)

2. Is the manuscript technically sound, and do the data support the conclusions?

Reviewer #1: Yes

Reviewer #2: Yes

3. Has the statistical analysis been performed appropriately and rigorously? 

Reviewer #1: Yes

Reviewer #2: Yes

4. Have the authors made all data underlying the findings in their manuscript fully available?

Reviewer #1: No

Reviewer #2: Yes

5. Is the manuscript presented in an intelligible fashion and written in standard English?

Reviewer #1: Yes

Reviewer #2: Yes

6. Review Comments to the Author

Reviewer #1: The authors acknowledge some limitations, but this section should also highlight the retrospective nature of the study and how it may have influenced the findings. The small number of correctly filled-out diaries and potential biases (e.g., patients more engaged with their treatment may have been more likely to return their diaries) should be elaborated on.

Reviewer #2: *Did the insomniac group returned sleep diaries in both USD and LSD group or just LSD-insomniac group? I think this is not clear just by reading the abstract (lines 37 and 40 in the R1 version w/ track changes).

* To encourage light data in the sleep diaries, there is no need to underestimate the measurement of actimeters, since it is quantitative data which is actually more reliable than diaries (line 57 in the R1 version w/ track changes).

* Figure 1 a and b should have legends to explain them. Also, non-english terms should be replaced with english words. It is not clear what up and down arrows or filled circle etc mean for a person just looking at the figure.

* If figures are labeled as Fig.1a/b, they should be mentioned just like that at lines 99 and 104.

* Table 1, Table 3 and abbreviations say ‘UD’ which should be replaced with ‘USD’ to keep the consistency throughout the manuscript

* Line 289, ‘Study’ should be ’study’

7. PLOS authors have the option to publish the peer review history of their article (what does this mean?). If published, this will include your full peer review and any attached files.

Reviewer #1: No

Reviewer #2: No

---

## [Author Response · Author response to Decision Letter 2]

17 Dec 2024

Dear Editor,

We would like to thank you for your interest in our work. We also express our gratitude to the reviewers of this second round, for their valuable comments and suggestions. They have helped improve this manuscript.

Please find below our responses to their comments and questions.

Sincerely,

Reviewer #1 :

The authors acknowledge some limitations, but this section should also highlight the retrospective nature of the study and how it may have influenced the findings.

Thank you for your comment. We apologize for the misunderstanding. This was indeed a prospective study. An empty sleep diary was sent to each patient (randomly assigned to either USD or LSD), and they were asked to fill it out daily until their next consultation. Sleep diaries were collected at that time and continued to be collected until the end of the study. The study is indeed prospective, with the sleep diary variable being randomized. Data regarding diary completion were gathered progressively during follow-up and do not represent pre-existing data. We have added a sentence to the Methods section to clarify this point (p7, line 116-119).

 The small number of correctly filled-out diaries and potential biases (e.g., patients more engaged with their treatment may have been more likely to return their diaries) should be elaborated on.

We believe this point is addressed in the Discussion section. We focused on insomniac patients, who indeed completed the diaries more consistently than OSAS patients, for instance (see page 11). We have further elaborated on this point in the Discussion, specifically on line 304.

Reviewer #2:

*Did the insomniac group returned sleep diaries in both USD and LSD group or just LSD-insomniac group? I think this is not clear just by reading the abstract (lines 37 and 40 in the R1 version w/ track changes).

Both USD and LSD were returned. We have modified the abstract for greater precision.

* To encourage light data in the sleep diaries, there is no need to underestimate the measurement of actimeters, since it is quantitative data which is actually more reliable than diaries (line 57 in the R1 version w/ track changes).

Our intention was not to create this impression but rather to highlight the differences in accessibility between actimeters coupled with luxmeters. We have revised the Introduction section to make this clearer (p4, line 57).

* Figure 1 a and b should have legends to explain them. Also, non-english terms should be replaced with english words. It is not clear what up and down arrows or filled circle etc mean for a person just looking at the figure.

Thank you for pointing this out. We have updated the sleep diary to include an English version.

* If figures are labeled as Fig.1a/b, they should be mentioned just like that at lines 99 and 104.

* Table 1, Table 3 and abbreviations say ‘UD’ which should be replaced with ‘USD’ to keep the consistency throughout the manuscript

* Line 289, ‘Study’ should be ’study’

These items have been reviewed and corrected.

---

## [Decision Letter · Decision Letter 2]

5 Jan 2025

How do people feel about sleep diaries? 

Factors influencing sleep diaries completion with and without daily exposure to light

PONE-D-23-38179R2

Dear Dr. Chambe,

We’re pleased to inform you that your manuscript has been judged scientifically suitable for publication and will be formally accepted for publication once it meets all outstanding technical requirements.

Kind regards,

Julio Alejandro Henriques Castro da Costa

Academic Editor

PLOS ONE

Additional Editor Comments (optional):

Reviewers' comments:

Reviewer's Responses to Questions

**Comments to the Author**

1. If the authors have adequately addressed your comments raised in a previous round of review and you feel that this manuscript is now acceptable for publication, you may indicate that here to bypass the “Comments to the Author” section, enter your conflict of interest statement in the “Confidential to Editor” section, and submit your "Accept" recommendation.

Reviewer #2: All comments have been addressed

2. Is the manuscript technically sound, and do the data support the conclusions?

Reviewer #2: Yes

3. Has the statistical analysis been performed appropriately and rigorously? 

Reviewer #2: Yes

4. Have the authors made all data underlying the findings in their manuscript fully available?

Reviewer #2: Yes

5. Is the manuscript presented in an intelligible fashion and written in standard English?

Reviewer #2: Yes

6. Review Comments to the Author

Reviewer #2: Thank you for carefully addressing the comments, I think this manuscript is valuable, well-written and contributes to the field and should be published in PLOS One Journal. I think LSD-type diaries has a potential to be implemented in sleep studies.

7. PLOS authors have the option to publish the peer review history of their article (what does this mean?). If published, this will include your full peer review and any attached files.

Reviewer #2: No

---

## [Editor Report · Acceptance letter]

PONE-D-23-38179R2

PLOS ONE

Dear Dr. Chambe,

I'm pleased to inform you that your manuscript has been deemed suitable for publication in PLOS ONE. Congratulations! Your manuscript is now being handed over to our production team.

Kind regards,

on behalf of

Dr. Julio Alejandro Henriques Castro da Costa

Academic Editor

PLOS ONE